# CAUSAL-BASED ANALYSIS ON CREDIBILITY OF FEEDFORWARD NEURAL NETWORK

## ABSTRACT

Feedforward neural network (FNN) has been widely used in various fields. However, its credibility in some risk-sensitive fields remains questionable. The complex causal relations between neurons in the input layer is hard to be observed. These causal relations affect the credibility of FNN directly or indirectly. We transform FNN into a causal structure with different causal relations in the input layer. In order to analyze the causal structure accurately, we categorize it into three causal sub-structures based on different causal relations between input and output neurons. With different levels of intervention, we analyze the causal effect by calculating average treatment effect for each neuron in the input layer. We conduct experiments in the field of pediatric ophthalmology. The results demonstrate the validity of our causal-based analysis method.

## 1 INTRODUCTION

The black-box nature of deep learning models leads to a lack of authority and credibility in risk-sensitive fields. Numerous researchers have analyzed the possibility of increasing credibility from an interpretability perspective. Miller et al (Miller, 2019) argued that interpretability refers to the ability of a human to understand why the model made this decision over another. Kim et al (Kim et al., 2016) argued that the interpretability refers to the extent to whether can humans predict a model's next decision and Doshi-Velez et al (Doshi-Velez & Kim, 2017) suggested that explaining to humans in an understandable way could increase credibility. In linear regression (Haufe et al., 2014), expressions for weights and biases can show the magnitude of each feature's influence on the prediction as well as positive and negative correlations. DeepLIFT (Deep Learning Important FeaTures) (Shrikumar et al., 2017) broke down the contributions of all neurons in a neural network to each input variable by backpropagation to decompose the prediction of a neural network for a specific input. Individual Condition Expectation (ICE) (Goldstein et al., 2015) improved credibility by plotting the relations between individual single characteristic variables and predicted values. Nauta et al (Nauta et al., 2021) used classification models to find out information about the most important visual features to enhance the prototype, thus improving credibility. LIME (Local Interpretable Model-Agnostic Explanations) (Ribeiro et al., 2016) trained interpretable models to approximate individual predictions without interpreting the whole model.

Pearl et al (Pearl, 2018) argue that causal interpretability approaches can answer questions related to interpretability and Looveren et al. (Van Looveren & Klaise, 2021) proposed the use of prototypes to guide the generation of counterfactuals. Chattopadhyay et al (Chattopadhyay et al., 2019) proposed an abstraction approach for neural networks. In this paper, we introduce a causal-based analysis method to analyze the credibility of feedforward neural network (FNN) decision reasoning. We focus on the causal relations between neurons in the input and output layers and establish a causal structure to describe the complex causal relations therein. In applications of various fields, there may be causal relations between input neurons that people are not aware of, which directly leads to biased output results and affect the credibility and authority of neural network decisions. We divide the causal structure into three sub-structures based on the possible causal characteristics of the input layer neurons. For different sub-structures, we propose different causal analysis methods and assess the credibility of neural network from the causal perspective of the whole process.

**Contributions.** The contributions of our work are as follows:

• We present a full-flow analysis of the credibility of feedforward neural networks from a causal perspective.
• We unify the relations between neurons in the input and output layers into three sub-structures and do causal analysis for each of them.

## 2 PRELIMINARIES

**D-separation(Directional Separation):** We use D-separation (Hayduk et al., 2003) to judge conditional independence in a complex causal structure. A path in the causal structure consists of a series of nodes, and there exists a directed edge between neighboring nodes, which points to the next node in the sequence. A path p will be blocked by a set of nodes Z if and only if:

(1) p contains a chain junction $A \to B \to C$ or a fork junction $A \leftarrow B \to C$ and node B is in the set of nodes Z.

(2) p contains collision junction $A \to B \leftarrow C$ and neither collision node B nor its children nodes are in Z.

**Confounder:** A common confounder is the co-causation of the intervened variable and the outcome variable.

**Backdoor path:** Backdoor paths are defined as all paths between X and Y that begin with an arrow pointing to X. The existence of such paths may lead to erroneous inferences of causality. Blocking the backdoor paths realizes the deconfounding of X and Y.

**Backdoor criterion:** In a directed acyclic graph, given an ordered pair (X,Y), a set of variables Z is considered to satisfy the backdoor criterion (Bareinboim & Pearl, 2016) for X and Y when the following two conditions are satisfied:

(1) Z blocks all backdoor paths from X to Y, or Z d-separate all backdoor paths from X to Y.

(2) Z does not contain any descendant nodes of X

On the basis of the backdoor criterion, we are supposed to eliminate confounder. If the variable Z satisfies the backdoor criterion for X and Y, then the causal effect of X on Y can be calculated by Eq.1.

$$E(Y|do(X=1)) - E(Y|do(X=0)) = E_Z E(Y|X=1, Z) - E_Z E(Y|X=0, Z) \tag{1}$$

## 3 FNN TO CAUSAL SUB-STRUCTURES

In practice, there will be causal relations between input neurons that humans can not observe. In order to analyze the credibility of FNN more accurately, we consider complex causal relations between input neurons. In FNN, only the input and output layers can be explicitly observed, while the hidden layers cannot be effectively observed. To simplify our causal structure, we delete the hidden layers because we focus more on input and output layer neurons. We marked neurons with causal relationships between input layers in green, while independent neurons remained in blue in Fig(b) and (c). Fig.1(c) shows the causal structure we established.

For a feedforward neural network $FNN(l_1, l_2, ..., l_n)$ with $n$ layers, there is a corresponding causal structure $CS([l_1, l_2, ..., l_n], [f_1^*, f_2, ..., f_n])$, where $l_i$ denotes the set of neurons in layer $i$, $l_1$ denotes the input layer, and $l_n$ denotes the output layer. Corresponding to each $l_i$, $f_i$ denotes the set of causal functions of neurons in layer $i-1$ to neurons in layer $i$, and $f_1^*$ denotes the set of causal functions consisting of causal relations among neurons in input layer. After deleting the hidden layer, we have the following causal structure, where $f^*$ denotes the set of causal functions consisting of causal relations among neurons in input layer and $f'$ denotes the set of causal functions of neurons in input layer to neurons in output layer as shown in Fig1(c).

$$CS'([l_1, l_n], [f^*, f']) \tag{2}$$

In order to discover the causal relations between input neurons, we use the Peter-Clark(PC) algorithm (Spirtes et al., 2000) based on Degenerate Gaussian(DG) (Andrews et al., 2019), where DG is a method to test independence between neurons. The main flow of the causal discovery algorithm is as follows:

(1) Create a completed undirected graph consisting of all input layer neurons and test the inde-

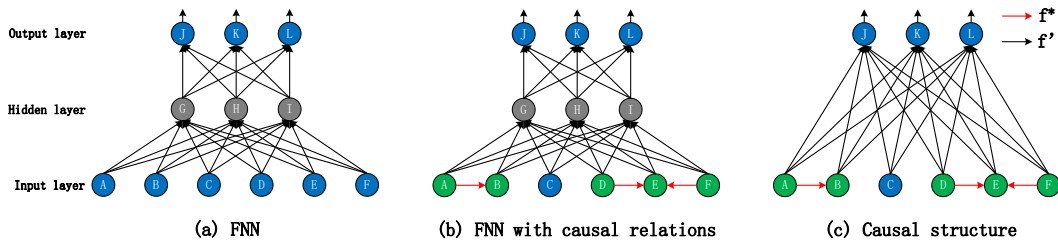

Figure 1: FNN to Causal structure

pendence among different neurons.

(1) Use D-separation to determine the direction of edges in the graph.

Further we categorized the causal structure into three different sub-structures based on the different causal relations we discovered: independent sub-structure, mediate sub-structure and confounding sub-structure. In independent sub-structure, neurons have no causal relations with other neurons. In mediate sub-structure, neuron $l_{1a}$ has a causal relation with neuron $l_{1b}$, neuron $l_{1b}$ becomes a mediator between neurons $l_{1a}$ and $l_{nc}$. In confounding sub-structure, neuron $l_{1b}$ is the co-causation of the intervened neuron $l_{1a}$ and the outcome neuron $l_{nc}$, so neuron $l_{1b}$ is a confounder between neurons $l_{1a}$ and $l_{nc}$. For different sub-structures, we propose different methods of causal analysis, which will be explained in detail in section 4.

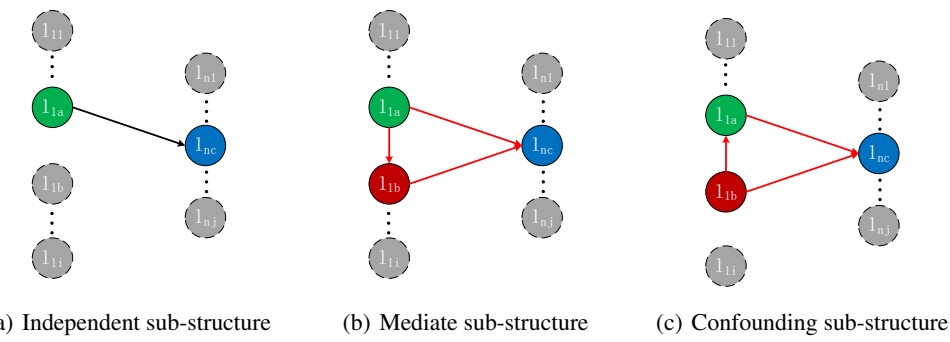

(a) Independent sub-structure     (b) Mediate sub-structure     (c) Confounding sub-structure

Figure 2: Three different sub-structures

# 4 CAUSAL ANALYSIS

## 4.1 INDEPENDENT SUB-STRUCTURE

We performed a causal analysis by imposing an intervention and assessing the causal effect of each input layer neuron on the output layer neurons. Causal paths refers to all the directed paths from the start neuron to the end neuron in the causal sub-structure. Based on all the identified causal paths, we calculate each average causal effect(ATE) of three sub-structures. For neurons in independent sub-structure, which have no causal relations with other neurons, we propose the following ATE calculation:

$$ATE^{y}_{do(l_{1i}=\alpha)} = E[y|do(l_{1i}=\alpha)] - baseline_{l_{1i}} \tag{3}$$

We define $ATE^{y}_{do(l_{1i}=\alpha)}$ as the causal effect of input neuron $x_i$ to output neuron $y$. When the domain is continuous, the gradient $\frac{\partial E[y|do(l_{1i}=\alpha)]}{\partial l_{1i}}$ is sometimes used to approximate the average causal effect

ATE. However, the gradient is subject to sensitivities that can cause the causal effect to be interfered with by other input layer neurons. In Eq.3, the ideal baseline is points on the decision boundary of the neural network at which predictions are neutral. However, Kindermans et al (Kindermans et al., 2019) showed that when a reference baseline is fixed to a specific value (e.g. zero vector), the method is not affine invariant. Thus the ATE of $l_{1i}$ on y can be written as the baseline value of $l_{1i}$ as $baseline_{l_{1i}} = E_{l_{1i}}[E_y[y|do(l_{1i} = \alpha)]]$.

Consider an output neuron y in the simplified $CS'([l_1, l_n], [f^*, f'])$. This causal function can be written as $y = f'_y(l_{11}, l_{12}, ..., l_{1k})$, where $l_{1i}$ is neuron i in the input layer and k is the number of input neurons. If a $do(l_{1i} = \alpha)$ operation is imposed on the network, the causal function is given by $f'_{y|do(l_{1i}=\alpha)}(l_{11}, ..., l_{1(i-1)}, \alpha, l_{1(i+1)}, ..., l_{1k})$. For brevity, remove the $do(l_{1i} = \alpha)$ subscript and simply refer to it as $f'_y$.

Let $\mu_j = E[l_{1j}|do(l_{1i} = \alpha)] \forall l_{1j} \in l_1$, $\mu = [\mu_1, \mu_2, ..., \mu_k]^T$ is a column vector. We assume $f'_y$ is a smooth causal function, the second-order Taylor's expansion of the causal function around the vector $\mu = [\mu_1, \mu_2, ..., \mu_k]^T$ is given by

$$f'_y(l_1) \approx f'_y(\mu) + \nabla^T f'_y(u)(l_1 - \mu) + \frac{1}{2}(l_1 - \mu)^T \nabla^2 f'_y(\mu)(l_1 - \mu) \tag{4}$$

Taking expectation on both sides:

$$E[f'_y(l_1)] \approx f'_y(\mu) + \frac{1}{2}Tr(\nabla^2 f'_y(\mu))E[(l_1 - \mu)(l_1 - \mu)^T] \tag{5}$$

Since $E[l_1|l_{1i} = \alpha] = \mu$, the first-order term $\nabla^T f'_y(u)(l_1 - \mu)$ disappears. It is now only necessary to calculate the vector $\mu$ and $E[(l_1 - \mu)(l_1 - \mu)^T|do(l_{1i} = \alpha)]$ to calculate Eq.8 (Chattopadhyay et al., 2019).

We propose Algorithm 1 to give causal analysis of input layer neurons on output layer neurons in independent sub-structure. In the interval from $low^i$ to $high^i$ of intervention $\alpha$, we give the value to $\alpha$ $num$ times uniformly. $Cov$ denotes the covariance matrix. Based on the causal function f, we input neuron x in input layer, neuron y in output layer and vector $\mu$. The output is an array of the causal effect corresponding to different $\alpha$.

---

**Algorithm 1** ATE on independent sub-structure

---

**Input:** $y, l_{1i}, [low^i, high^i], num, \mu, Cov, f()$
**Output:** $e[]$
1: $Cov[l_{1i}][:] := 0; Cov[:][l_{1i}] := 0$
2: $intervention\_expection := []; \alpha = low^i$
3: **while** $\alpha \leqslant high^i$ **do**
4: $\quad \mu[i] = \alpha$
5: $\quad intervention\_expection.append(f(\mu) + \frac{1}{2}trace(matmul(\nabla^2 f(\mu), Cov)))$
6: $\quad \alpha = \alpha + \frac{high^i - low^i}{num}$
7: $\quad add \quad E[y|do(l_{1i} = \alpha)] \quad to \quad array \quad e[]$
8: **end while**
9: **return** $e[]$

---

### 4.2 MEDIATE SUB-STRUCTURE

Because of the presence of mediator in mediate sub-structure, we can not directly calculate the causal effect using Algorithm 1. Consider an output neuron $l_{n_c}$ in the simplified $CS'([l_1, l_n], [f^*, f'])$ as shown in Fig.3(b). The intervened input layer neuron is assumed to be $l_{1a}$ and there is no confounder between $l_{1a}$ and $l_{n_1}$. It is necessary to identify all causal paths between $l_{1a}$ and $l_{nc}$. In Fig.3(b), there are two causal paths of $l_{1a}$ to $l_{nc}$, denoted as $P_1 = l_{1a} \rightarrow l_{nc}$ and $P_2 = l_{1a} \rightarrow l_{1b} \rightarrow l_{nc}$. The $P_2$ path is a chain junction, where $l_{1b}$ neuron can be considered as a mediator. After intervening on neuron $l_{1a}$, the distribution of neuron $l_{1b}$ in the input layer is also affected due to the presence of the $l_{1a} \rightarrow l_{1b}$ path. Similarly, this effect is reflected in the $l_{1b} \rightarrow l_{nc}$ path. Thus, the total causal effect of $l_{1a}$ on $l_{nc}$ should be the sum of the causal effect calculated on these two causal paths. When intervening on $l_{1a}$, the value of $l_{1b}$ changes accordingly because of the causal relation between $l_{1a}$

and $l_{1b}$. $l_{1a}$'s average causal effect on $l_{1b}$ is calculated as in Eq.6.

$$ATE = E[l_{1b}|do(l_{1a} = x_i)] - E[l_{1b}|l_{1a}] \tag{6}$$

In addition, we conduct three validation experiments between neurons $l_{1a}$ and $l_{1b}$ in order to assess the reliability of ATE and define the error rate as the Eq.7.

(1) Bootstrap Validation (BV): The estimated causal effect should not change significantly when replacing a given dataset with a sample from the same dataset.

(2) Add Random Common Cause (ARCC): The algorithm adds random variables to the data set as confounders to determine the causal effect. The correct causal effect should not change much with the addition of random confounders. Therefore, the better the algorithm works, the smaller the difference between the new causal effect estimate and the original estimate.

(3) Data Subsets Validation (DSV): The algorithm randomly selects a subset of the original data as the new dataset, removes some of the data and performs causal effect estimation. If the assumptions are correct, the causal effect will not change much.

$$ERR_{BV/ARCC/DSV} = |\frac{NewEffect - EstimatedEffect}{EstimatedEffect}| \tag{7}$$

After calculating the average causal effect between $l_{1a}$ and $l_{1b}$, $\mu_{1b}$ as the expectation of $l_{1b}$ after intervention $l_{1a}$ can be obtained. We still use Eq.5 after modify the vector $\mu$ and covariance matrix $Cov$ to calculate ATE between $l_{1a}$ and $l_{nc}$.

## 4.3 CONFOUNDING SUB-STRUCTURE

In real life, confounders are often interspersed between causal relations. Consider a common confounding sub-structure, as shown in Fig.3(c). The causal path from neuron $l_{1a}$ to neuron $l_{nc}$ is only $P_1 = l_{1a} \rightarrow l_{nc}$. Since $l_{1b}$ also points to $l_{nc}$, the intervention on $l_{1a}$ will be affected by $l_{1b}$ so that bias appears on $l_{1a}$. Neural $l_{1b}$ is a confounder that prevent us from using the causal effect calculations in Eq.5. The calculation of ATE due to the presence of confounder will be a two-step process:

(1) Find all the backdoor paths from the causal structure.

(2) Identify covariates needed to block backdoor paths based on backdoor criterion

(3) Give causal analysis of the covariates identified in the second step along with neurons of input layer and output layer in the confounding sub-structure.

We adopt the Domain Adaptive Meta-Learning algorithm (Battocchi et al., 2019) for the third step. We use propensity score to adjust the covariate Z, which is denoted by $e(z)$ as shown in Eq.10. The propensity measures the propensity of receiving the intervention in a given condition.

$$e(z) = P[X = 1|Z = z] \tag{8}$$

We calculate the causal effect of confounding sub-structure as follows:

(1) Defining $g(z)$ as an estimate of the propensity score $e(z)$

$$g(z) = \hat{e}(z) \tag{9}$$

(2) Define the resultant functions whose interventions are 0 and 1: $\mu_0(x) = E[Y(0)|X = x]$ and $\mu_1(x) = E[Y(1)|X = x]$. Therefore, the algorithm combines the propensity score to estimate both outcomes, defining the estimate:

$$\hat{\mu}_0 = M_1(Y^0 \sim X^0, weight = \frac{g(X_0)}{1 - g(X_0)}) \tag{10}$$

$$\hat{\mu}_1 = M_2(Y^1 \sim X^1, weight = \frac{g(X_1)}{1 - g(X_1)}) \tag{11}$$

where M denotes the machine learning model, $M(Y \sim X)$ denotes the estimation of $E[Y|X = x]$, $\hat{\mu}_0$ and $\hat{\mu}_1$ denote their corresponding prediction results.

(3) Calculate causal effect for the intervention and control groups for two steps. First, causal effect is calculated for individuals in the intervention group using the predictive model for the control group to estimate their causal effect. Second, individuals in the control group are estimated using

the predictive model for the intervention group and their causal effect is calculated. As shown in Eq.12 and Eq.13.

$$\hat{D}_i^1 = Y_i^1 - \hat{\mu}_0(X_i^1) \tag{12}$$

$$\hat{D}_i^0 = \hat{\mu}_1(X_i^0) - Y_i^0 \tag{13}$$

where $Y_i^1$, $Y_i^0$ denote the outcomes of the intervention and control groups, respectively, and $X_i^1$, $X_i^0$ are the corresponding eigenvectors, also called covariates. This step involves calculating the causal effect for the control and intervention groups separately to better address the imbalance between these two groups. By doing this, the causal effect can be estimated more accurately.

(4) Calculate the final causal effect. The outcome is estimated by the prediction model to obtain the causal effect estimate. The formula is defined as follows:

$$\tau(x) = E[\hat{D}|X = x] \tag{14}$$

$$\hat{\tau} = M_3(\hat{D}^0 \cup \hat{D}^1 \sim X^0 \cup X^1) \tag{15}$$

Where $\hat{D}^0 \cup \hat{D}^1$ denotes the dataset joining of the causal effect obtained for the intervention and control groups. $X^0 \cup X^1$ denotes the dataset joining for the covariate. $M(\hat{D} \sim X)$ denotes the estimation of $E[\hat{D}|X = x]$. Taken together, the final causal effect estimate $\hat{\tau}$ is obtained. We also conduct three validation experiments between neurons $l_{1a}$ and $l_{nc}$.

## 5 EXPERIMENTS

In order to analyze the credibility of FNN decision reasoning, we construct a FNN from a medical problem and calculate ATE based on causal relations. The medical field has a high demand for credibility. Therefore, our experiments aim to verify the causal analysis method proposed in this paper, which analyze the credibility of FNN in practical problems in the medical field. We conduct experiments using a pediatric ophthalmology dataset from a city $X$ in China, which is based on annual follow-up surveys from 2011 to 2017, covering a variety of aspects, including individual activities, parental genetics, diet and so on. We first conduct causal discovery experiment of input layer neurons based on our causal structure. Based on the results of the causal discovery experiment, we conduct causal analysis experiment of three different sub-structures, while we add validation experiments on the second and third sub-structure to assess the reliability of causal effect estimates. We selected 16 core variables as input neurons, whose names, meanings, types and ranges are listed in the Appendix.

### 5.1 CAUSAL DISCOVERY OF INPUT LAYER

To discover causal relations between neurons in the input layer, we conduct the causal discovery experiment. Our experiment performs causal discovery based on a PC algorithm with Degenerate Gaussian for the obtained 16 variables of the pediatric myopia dataset. We use the TETRAD (Ramsey et al., 2018) library in constructing the causal graph. The final causal graph constructed by the experiment is shown in Fig.4. The experiment finds causal relations among a total of 10 variables out of 16, constructing a total of 15 edges. We discover DAI, GENDER, JTR, YTR, K1, and K2 affect both AL and RA, HEIGHT and REDM only affect AL, and AL affects RA.

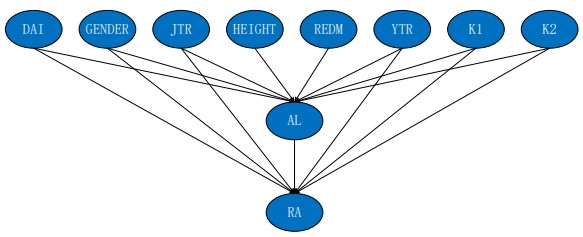

Figure 3: Causal discovery of pediatric ophthalmology dataset

## 5.2 CAUSAL ANALYSIS OF INDEPENDENT SUB-STRUCTURE

Based on the causal discovery experiment in the previous section, we summarize and generalize three different sub-structures. In this section we conduct a causal analysis experiment on independent sub-structures. For PULSE, the experiment results are shown in Fig.5, other results are shown in the Appendix. According to Fig.5, as the intervention value increases, PULSE tends to increase on output neuron labeled No, while PULSE tends to decrease on output neuron labeled Yes. Several studies have suggested that high pulse counts may be negatively associated with myopia (SHIH et al., 1991). That is, FNN's reasoning on PULSE is credible.

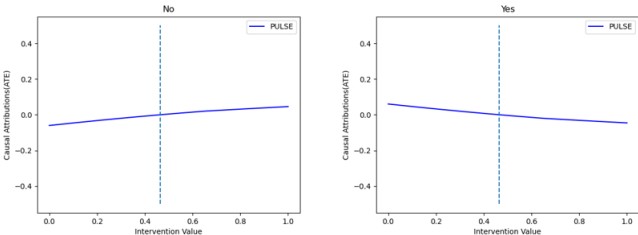

Figure 4: Causal analysis results of PULSE

## 5.3 CAUSAL ANALYSIS OF MEDIATE SUB-STRUCTURE

We use the method mentioned in Section 4.2 to assess the causal effect of the intervened neuron on the mediator. The final causal effect diagram is shown in Fig.6, where positive causal effects are labeled as solid lines and negative causal effects are labeled as dashed lines. The numbers on the arrows in the figure are the values of the causal effect between two neurons. The results of three validation experiments between neurons $l_{1a}$ and $l_{1b}$ are shown in Table 1. The error rates obtained in three validation experiments are less than 9%, and there is no situation in which the error rates of all three validation experiments are higher than 1%. This indicates that the causal effect estimates obtained are reliable.

After obtaining reliable results for the assessment of causal effects between variables, we conduct the causal analysis experiment. We only list the experiment results for HEIGHT as shown in Fig.7, and other results are shown in the Appendix. A study published in 2016 in the journal PLoS One found that taller people were more likely to be myopic among more than 60,000 respondents from several European countries (Hamade et al., 2016). Therefore, it can be concluded that FNN's reasoning on HEIGHT is credible.

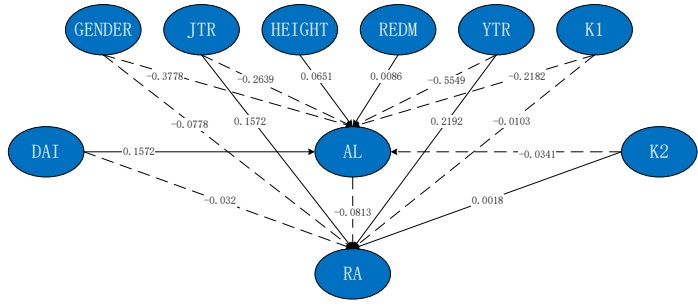

Figure 5: Causal effect of the pediatric ophthalmology dataset

## 5.4 CAUSAL ANALYSIS OF CONFOUNDING SUB-STRUCTURE

For confounding sub-structure, we adopt the Domain Adaptive Meta-Learning algorithm, but the meta-learning model requires the inputs to be discrete variables. We use equal-width discretization to discretize AL and RA and assess the average causal effect values between the discrete values as

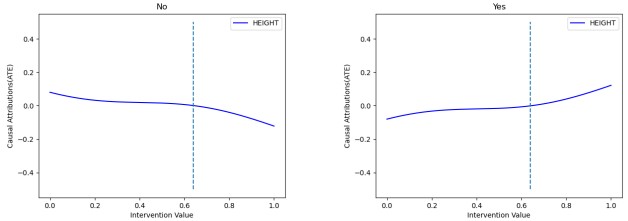

Figure 6: Causal analysis results of HEIGHT

Table 1: Validation experiments for variables with causal relations

| INTERVENTION | RESULT | BV% | ARCC% | DSV% |
|---|---|---|---|---|
| DAI | AL | 0.6291 | 0.0297 | 1.2914 |
| DAI | RA | 6.2595 | 0.0275 | 0.2702 |
| GENDER | AL | 0.9775 | 0.0130 | 1.6287 |
| GENDER | RA | 1.1297 | 0.0192 | 1.1078 |
| JTR | AL | 2.0231 | 0.0037 | 6.7741 |
| JTR | RA | 6.8549 | 0.0030 | 2.8822 |
| HEIGHT | AL | 0.9340 | 0.0096 | 0.9139 |
| REDM | AL | 1.1528 | 0.0542 | 1.2745 |
| YTR | AL | 2.6925 | 0.0013 | 1.0469 |
| YTR | RA | 2.5916 | 0.0003 | 1.1852 |
| K1 | AL | 0.5381 | 0.0026 | 0.0575 |
| K1 | RA | 1.3375 | 0.0214 | 2.2884 |
| K2 | AL | 7.6269 | 0.0109 | 8.6001 |
| K2 | RA | 3.2185 | 0.1668 | 0.981 |
| AL | RA | 3.4753 | 0.0234 | 3.4161 |

shown in Fig.8. The results of three validation experiments between input layer neurons AL, RA and output layer neurons No, Yes are shown in Table 2.

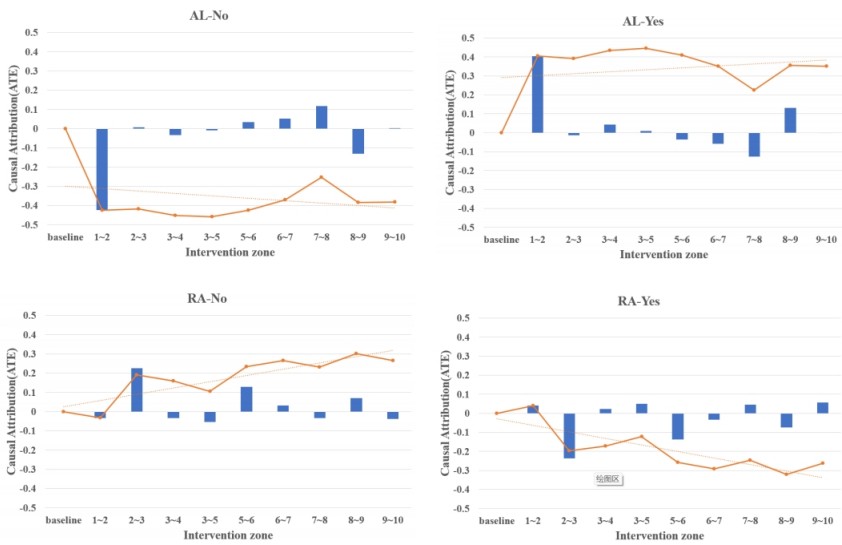

Figure 7: Causal analysis results of AL and RA

Table 2: Validation experiments for AL, RA, No and Yes

| INTERVENTION | RESULT | BV% | ARCC% | DSV% |
|---|---|---|---|---|
| AL | No | 21.46 | 4.48 | 32.03 |
| AL | Yes | 22.86 | 6.45 | 34.00 |
| RA | No | 1.11 | 4.48 | 7.54 |
| RA | Yes | 1.01 | 4.47 | 3.35 |

## 5.5 ABLATION EXPERIMENTS

To demonstrate the effectiveness of our causal analysis method. We also conducted ablation experiments. We removed the causal relationship between input layer neurons in section 3, restoring independence between each neuron. We conducted another causal analysis on each neuron in Fig.5 based on the method in Section 4.1. Due to space limitations, we have only listed the experiment results of HEIGHT and K1, while we have included the experimental results of other input layer neurons in Appendix C. From the experiment results, it can be seen that the curve becomes smoother without considering the complex causal relationships between input layer neurons. The credibility analysis of neural network will become more difficult.

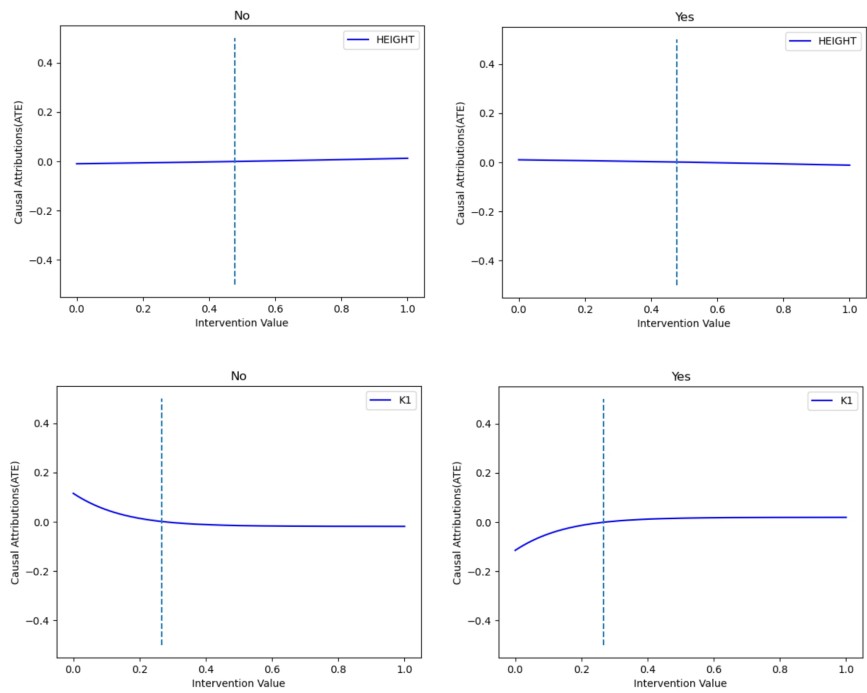

Figure 8: Ablation experiment results of HEIGHT and K1

## 6 CONCLUSION

In this paper, we propose a credibility analysis method for FNN from a causal perspective. We transform FNN into three different causal sub-structures to calculate its causal effect. We conduct full-flow experiments on different sub-structures from the discovery of causal relations to the calculation of causal effect. At the same time, we conduct validation experiments on different causal effects and prove their accuracy. The results demonstrate the validity of our method of causal-based analysis on FNN. Our work provides a new idea for the application and research of deep learning in risk-sensitive fields.

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

# A  VARIABLES OF THE PEDIATRIC OPHTHALMOLOGY DATASET

Table 3: Variables of the pediatric ophthalmology dataset

| VARIABLE | VARIABLE MEANING | DATA TYPE | DATA RANGE |
|---|---|---|---|
| RA | Post-dilated refraction | continuous variable | -5.4-8.7 |
| AL | eye shaft length | continuous variable | 20-34 |
| JG | Total proximity workload | continuous variable | 0-102 |
| YW | Amount of long-distance outdoor activity | continuous variable | 0-57 |
| JTR | Right eye accommodation at close range | continuous variable | -5.1-2.2 |
| YTR | Right eye accommodation at far range | continuous variable | -6.6-4.4 |
| HEIGHT | height | continuous variable | 97-143 |
| PULSE | pulse rate | continuous variable | 52-140 |
| GENDER | gender | binary variable | 1,2 |
| COLA | Frequency of carbonated beverages | discrete variable | 1-5 |
| EGG | Frequency of eggs | discrete variable | 1-4 |
| REDM | Frequency of red meat | discrete variable | 1-5 |
| WHIM | Frequency of white meat | discrete variable | 1-5 |
| DAI | Number of parents wearing glasses | discrete variable | 0-2 |
| K1 | corneal curvature in both eyes | continuous variable | 38-71 |
| K2 | corneal curvature in both eyes | continuous variable | 38-183 |

# B  CAUSAL ANALYSIS RESULTS OF OTHER VARIABLES

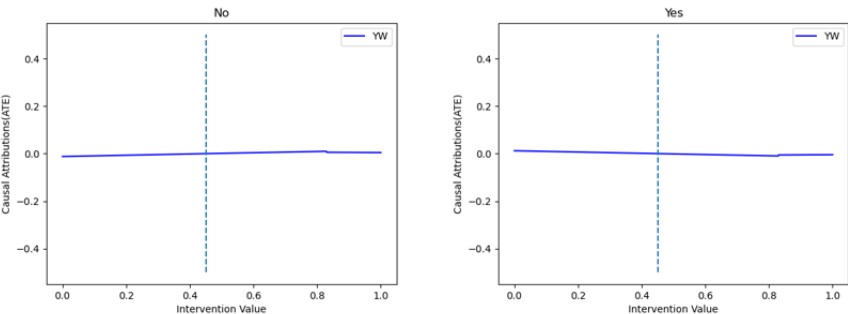

Figure 9: Causal analysis results of YW

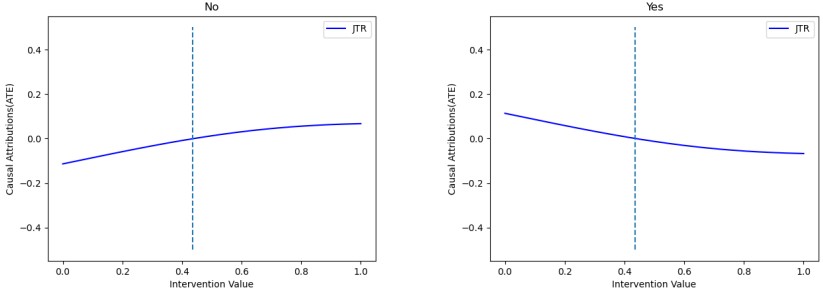

Figure 10: Causal analysis results of JTR

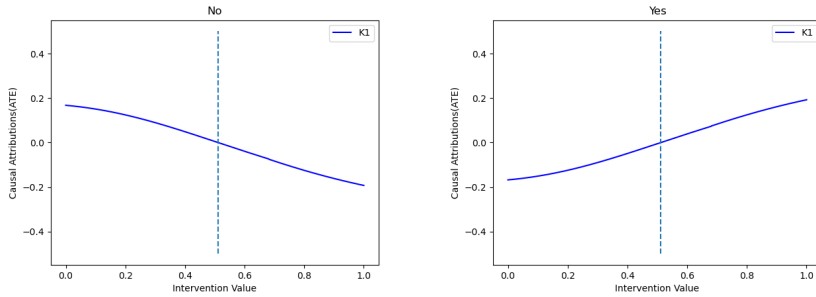

Figure 11: Causal analysis results of K1

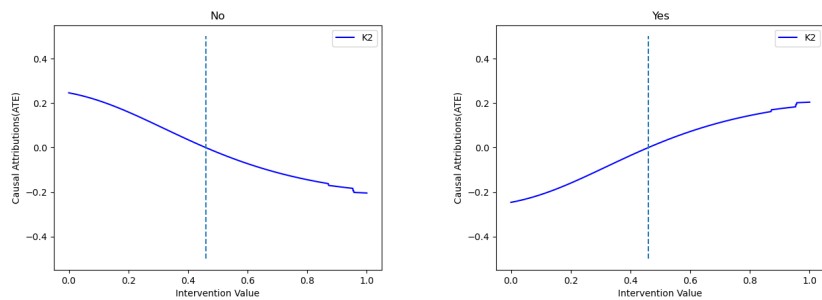

Figure 12: Causal analysis results of K2

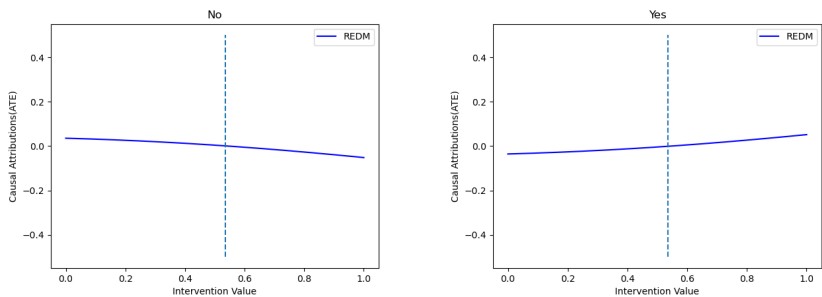

Figure 13: Causal analysis results of REDM

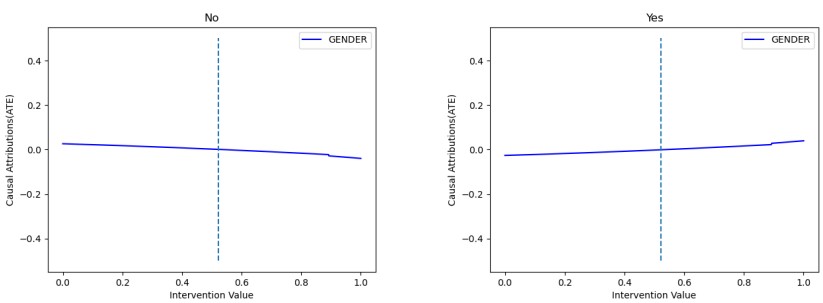

Figure 14: Causal analysis results of GENDER

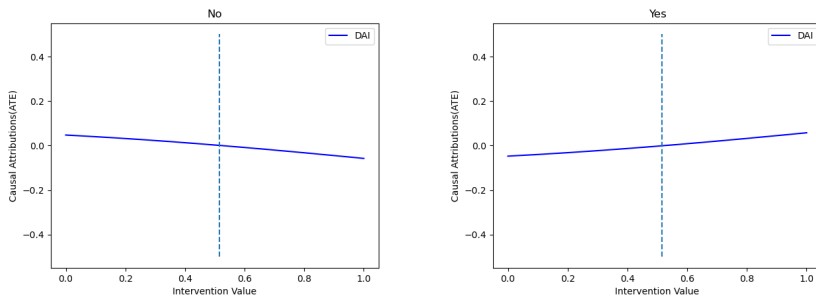

Figure 15: Causal analysis results of DAI

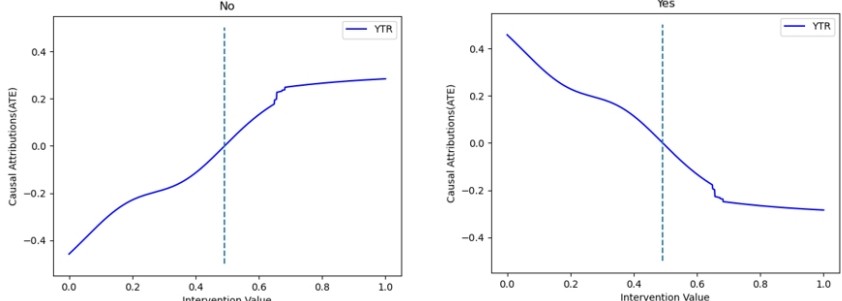

Figure 16: Causal analysis results of YTR

# C ABLATION EXPERIMENT RESULTS OF OTHER VARIABLES

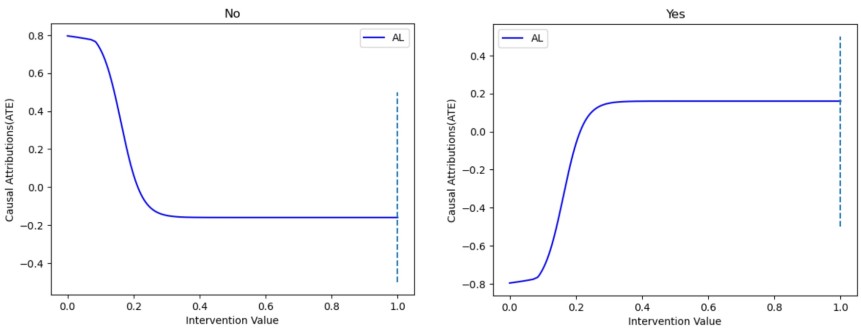

Figure 17: Ablation experiment results of AL

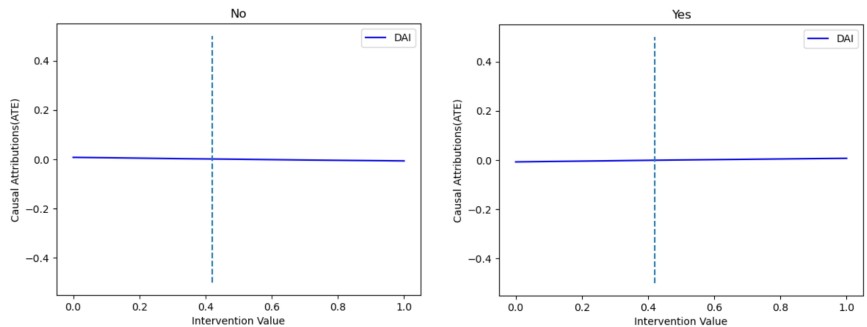

Figure 18: Ablation experiment results of DAI

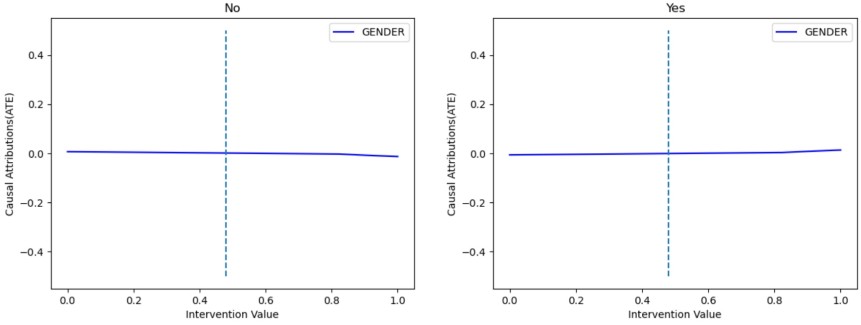

Figure 19: Ablation experiment results of GENDER

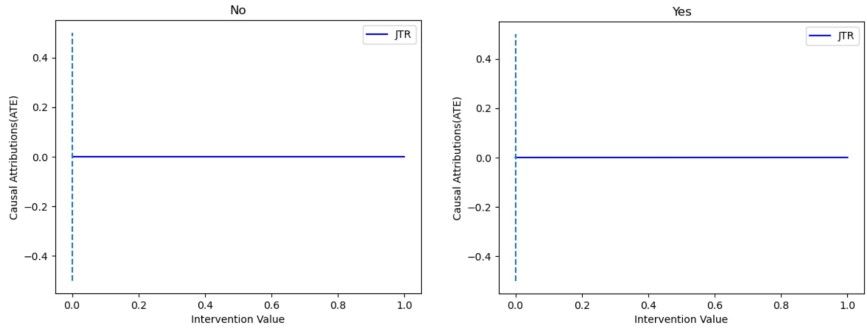

Figure 20: Ablation experiment results of JTR

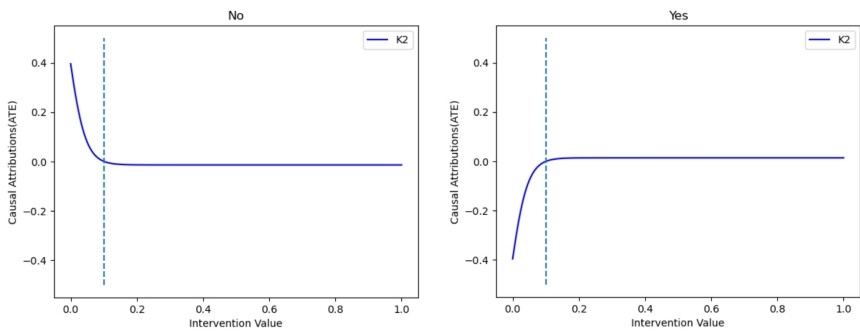

Figure 21: Ablation experiment results of K2

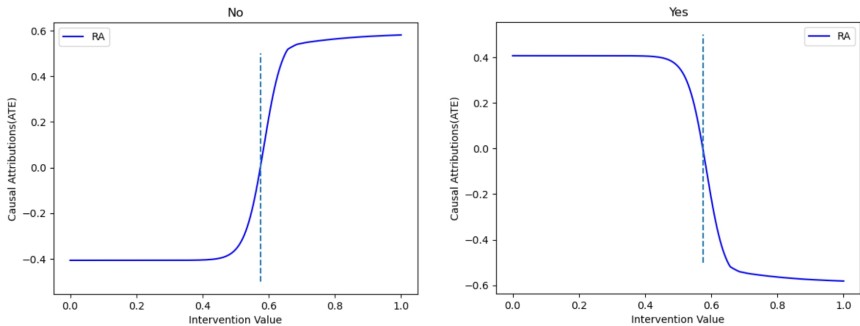

Figure 22: Ablation experiment results of RA

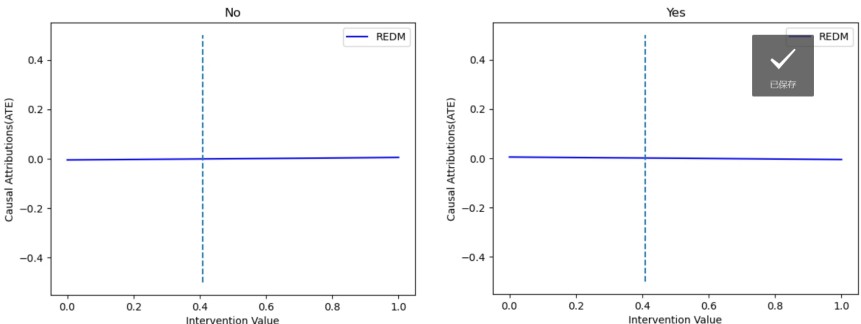

Figure 23: Ablation experiment results of REDM

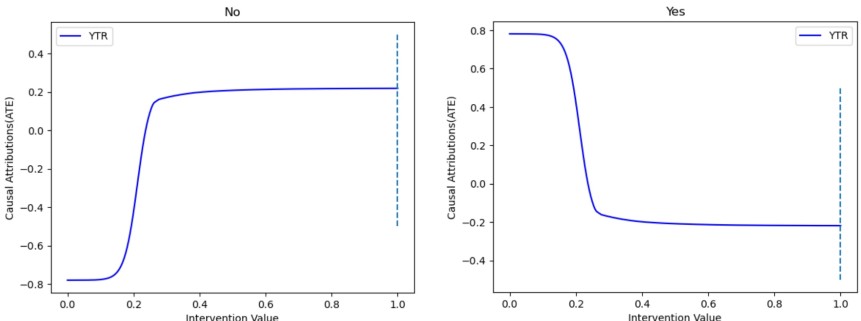

Figure 24: Ablation experiment results of YTR

