# OpenReview forum: "Causal-based Analysis on Credibility of Feedforward Neural Network"
_ICLR.cc/2024/Conference — ICLR 2024 Conference Withdrawn Submission_

### Official Review · Reviewer_wRkd · 2023-10-26

**Soundness:** 2 fair
**Presentation:** 2 fair
**Contribution:** 3 good
**Rating:** 5
**Confidence:** 4

**Summary:**

The authors propose to address the problem of feedforward neural network (FNN)’s credibility by discovering and quantifying the causal relationships in the input and output layers of FNN. To estimate the causal effect of intervening in an input neuron, the authors categorize the causal structure of the input layer into 3 different substructures and calculate the corresponding average treatment effects. Experiments are conducted in a medical setting to demonstrate the effectiveness of the proposed methodology.

**Strengths:**

•	The issue of FNN’s credibility is important and holds practical value in various settings, including medical applications.

•	The authors provide a sufficient introduction to the background knowledge of causal inference and feedforward neural networks before presenting the main methodology.

•	The paper is well organized.

**Weaknesses:**

•	The definitions and illustrations of “causal substructure” are ambiguous. The authors give explanations using individual neurons $l_{1a}$, $l_{1b}$, and $l_{nc}$, but this gives me an intuition that cases (b) and (c) in Figure 3 need not to be treated separately. For example, if I **swap $l_{1a}$ and $l_{1b}$ in case (c) and intervene on $l_{1b}$**, then case (c) is just equivalent to case (b). In other words, every time I encounter a confounding substructure, I can just swap the order of input features and intervene on the same neuron to transform it into a mediation substructure. Is that right? Please correct me if I misunderstood anything.

•	The notations in Section 4 are very confusing as the authors use $l_{1a}, l_{1b}$ and $x_{i}$ interchangeably to represent input neurons. My understanding is that $x_{i}$ represents some value that the neuron $l_{1i}$ can take. If so, then the expressions $E[\cdot|x_{i} = \alpha]$ and $E[\cdot|do(x_{i} = \alpha)]$ again become very confusing.

Minor suggestions:

•	For the definition of d-separation, it might be better to note that the set $Z$ is measured/observed so that the remaining unobserved variables can be described by distributions conditional on $Z$.

•	To make the illustration clear, it might be better to clearly mark $f^{*}$ and $f’$ in Figure 1(c).

•	Equations 7-9 appear to have similar expressions but are applied in different contexts. The authors might consider condensing this section by providing more detailed context and presenting a single equation.

**Questions:**

•	Could you elaborate more on the “causal substructure” as I mentioned in the first bullet point in the Weaknesses section?

•	Right above Equation 4, you assume that $f’_{y}$ is smooth. Does this mean you assume there are no non-smooth activation functions in the FNN (e.g., ReLU whose derivative does not exist at zero)?

•	In Section 3, why do you choose threshold values of 1% and 9% for the error rates?

---

> ### Author Response · Authors · 2023-11-22
>
> We thank the reviewer for your positive and constructive feedbacks. We have modified the annotations in Section 4  to make our expression clearer. To make the illustration clear, we mark $f^*$ and $f’$ in Figure 1(c). We also have condensed Eq.7-9 by presenting a single equation. Regarding the issue you mentioned in Figure 3, we analyze the three figures in Figure 3 with neuron l1a as the protagonist, so in Figures 3b and 3c, neuron l1a is in different causal situations. On the other hand, your viewpoint is also very correct. If we exchange the positions between the two input layer neurons l1a and l1b in Figures 3b and 3c, their situations can also be classified into one category. So from another perspective, it can also be divided into two types of substructures, but this is not the original intention we want to showcase. After observing the experimental results, we choose threshold values of 1% and 9%, which may lack some rigor. In addition, we supplemented the ablation experiment, and without considering the causal relationship between input layer neurons. It can be seen that our ATE slope has significantly decreased, making the analysis of neural network credibility more difficult. We apologize for not clarifying all questions given the limited space and many reviews. We will fix all typos and add missing references in the next revision.

---

> > ### Comment · Reviewer_wRkd · 2023-11-23
> >
> > I thank the authors for their response and effort to update the manuscript. I will keep my score unchanged for now as I think my concerns are only partially addressed based on this response.

---

### Official Review · Reviewer_3wbn · 2023-10-29

**Soundness:** 3 good
**Presentation:** 2 fair
**Contribution:** 2 fair
**Rating:** 3
**Confidence:** 2

**Summary:**

To enhance the credibility of feedforward neural networks (FNNs), the paper transforms the FNN model into a causal structure with different causal relationships between the nodes of the input layer. Based on three categories of causal structures in the input layer, the causal effect of the potentials interventions are calculated. The proposed method is evaluated and validated with experiments in the field of pediatric ophthalmology.

**Strengths:**

1. The problem studied is fundamental and important.

2. The idea is natural.

**Weaknesses:**

The validation is not very convincing. To validate the effectivensss of the method, I think randomized controlled trials to establish the causal relationship between nodes of the input layer and that between the input and output nodes. The experimental results should serve as the ground truth to evaluate the method.

**Questions:**

How can we design randomized controlled trials to validate the effectivenss of the proposed method?

---

> ### Author Response · Authors · 2023-11-22
>
> Thanks for the helpful comments!  We believe that our method can be further expanded by combining deep neural networks. We supplemented the ablation experiment, and without considering the causal relationship between input layer neurons. It can be seen that our ATE slope has significantly decreased, making the analysis of neural network credibility more difficult. We will address all remaining minor suggestions in the final revision.

---

### Official Review · Reviewer_EaUh · 2023-10-30

**Soundness:** 1 poor
**Presentation:** 2 fair
**Contribution:** 2 fair
**Rating:** 3
**Confidence:** 4

**Summary:**

The authors aim to use causal structure learning to help establish the credibility of causal interpretations of neural networks.

**Strengths:**

I like the idea of using causal structure learning algorithms to help identify or verify causal structures in neural networks. Finding a good relationship between causal structure learning and NN learning would really help to clarify the NN field focused on giving causal interpretations of NNs.

**Weaknesses:**

The paper could be focused a bit more, I think, by a couple of rounds of rewriting. Some specific points:

1. The flowchart in Figure 2 did not completely make sense. For instance, it looks like adding causal is independent of FNN, and it also looks like deleting hidden layers is an effect that has no downstream effects. Neither of these makes sense to me, given the discussion.

2. Rhetorically, the first paragraph is somewhat disorganized and could potentially be broken into a few paragraphs for clarity.

**Questions:**

I have some questions.

1.	Is Degenerate Gaussian being used because the variables are mixed continuous/discrete? It seems that a neural net is being used as well, which suggests that, in that case, the variables are being treated as continuous--unless there is some encoding that I haven’t seen here. Figure 6 suggests that all of the variables are being treated as continuous, in which case the Degenerate Gaussian test is unnecessary; one could simply use (in Tetrad) the Fisher Z test (i.e., conditional correlation).

2.	Also, for Figure 4, is tiered knowledge in Tetrad being assumed here, with edges forbidden in Tier 0? Otherwise, why are there no edges in Tier 1? This should be clarified. It seems unreasonable in Tetrad for edges concurrent in a tier not to exist, unless they are explicitly forbidden.

3.	Suppose knowledge is being assumed with 3 tiers as depicted with edges forbidden in Tier 0. In that case, all possible edges between tiers are represented here, which suggests that no edge has been ruled out. Does this make sense? Is this observed even if the algorithm, score, or test are varied, with variations in parameters? (I.e., it seems one needs a sensitivity test here.)

4.	Also, it was unclear to me why the neurons in the hidden layer were left out of account if data for those is available.

5.	Another issue is that this neural net consists of just three layers. Can this analysis extend to deep neural nets, a subject of considerable interest in recent years?

6.	The flowchart in Figure 2 did not completely make sense. For instance, it looks like adding causal is independent of FNN, and it also looks like deleting hidden layers is an effect that has no downstream effects. Neither of these makes sense to me, given the discussion.

7. The NN considered consists of just three layers. Can this analysis extend to deep neural nets, a subject of considerable interest in recent years?

---

> ### Author Response · Authors · 2023-11-22
>
> Thanks for the helpful comments!  We believe that our method can be further expanded by combining deep neural networks. We supplemented the ablation experiment, and without considering the causal relationship between input layer neurons. It can be seen that our ATE slope has significantly decreased, making the analysis of neural network credibility more difficult. The causal diagram in Figure 4 is a comprehensive causal diagram obtained by subtracting unnecessary edges based on the clinical experience of professional doctors after performing causal discovery algorithms on the data. To simplify our causal structure, we delete the hidden layers because we focus more on input and output layer neurons. We focus on the causal relationship and effects between input layer neurons and output layer neurons, and further analyze the credibility of neural network decisions. To simplify the analysis process, we have omitted hidden layer neurons. We will thoroughly check and fix grammatical errors in the final submission.

---

### Official Review · Reviewer_HFVt · 2023-10-31

**Soundness:** 2 fair
**Presentation:** 2 fair
**Contribution:** 2 fair
**Rating:** 5
**Confidence:** 2

**Summary:**

This paper proposes a credibility analysis method for FNN from a causal perspective. It transforms FNN into three different causal sub-structures to calculate its causal effect. The authors of this paper conducted full-flow experiments on different sub-structures from the discovery of causal relations to the calculation of causal effect. At the same time, it conducts validation experiments on different causal effects and proves their accuracy. The results demonstrate the validity of the method of causal-based analysis on FNN. This work provides a new idea for applying and researching deep learning in risk-sensitive fields.

**Strengths:**

1.	This paper effectively conducts a full-flow analysis of the credibility of feedforward neural networks from a causal perspective.
2.	This paper unifies the relations between neurons in the input and output layers into three sub-structures and do causal analysis for each of them.
3.	The experimental results in the field of pediatric ophthalmology demonstrate the validity of the proposed causal-based analysis method.

**Weaknesses:**

1.	The motivation of this paper seems unclear. It would be beneficial to provide a clear explanation of its underlying principles that establish the credibility of causal analysis methods in risk-sensitive domains.
2.	In Figure 1, the subfigures (a), (b), and (c) depict causal structures, but the explanatory text and the symbols are not clearly defined.
3.	In Figure 2, the authors divide the causal structure into three distinct substructures to analyze the credibility of the feedforward neural network. However, there is a lack of sufficient description regarding the reasons for dividing the causal structure into three separate substructures, as well as the relationships and differences between each structure shown in Figure 3.
4.	The confusion substructure proposed in Figure 3(c) lacks a clear explanation of the elements depicted in the figure, such as what each element represents. Additionally, the paper does not utilize mainstream causal structure models and does not provide specific elements of the causal graph. Furthermore, there is a lack of comparative description between Figure 3(c) and Figure 3(b).
5.	The experiments are limited by only using one dataset. The paper lacks specific case studies and ablation experiments, making it difficult to verify the robustness and generalization of the algorithm. Additionally, there is no comparison with the state-of-the-art methods.

**Questions:**

1. The effectiveness of this method was only validated in the medical field during the experiment. Could the authors validate the method's ability to analyze credibility in multiple domains using more diverse datasets?
2. In Figure 1, there is a lack of text description and symbol interpretation.
[1] In Figure 1(a), it should be indicated in the figure what each layer represents, and labels should be assigned to each layer from bottom to top, such as "input layer", "hidden layer", and "output layer" for clarity.
[2] In Figure 1(b), why are A, B, D, E, F, and C represented in different colors? Do they have any specific meanings?
[3] In Figure 1(b) and (c), what do the paths AB and DEF respectively represent?
3. The work in this paper is based on causal analysis methods. However, it lacks a description of causal theory and related work in the field of deep learning, e.g.
[1] Liu Y, Wei Y S, Yan H, et al. Causal reasoning meets visual representation learning: A prospective study[J]. Machine Intelligence Research, 2022, 19(6): 485-511.
[2] Mao C, Cha A, Gupta A, et al. Generative interventions for causal learning[C]//Proceedings of the IEEE/CVF Conference on Computer Vision and Pattern Recognition. 2021: 3947-3956.
4. Regarding Figure 3, there is a need for a clear explanation of subfigures (b) and (c), including the meaning of each path and how confounding factors are introduced, such as llb serves a confounding factor between lla and lnc to bring the confounding effects. Additionally, it should be clarified which paths are identified as backdoor or frontdoor paths that require intervention.
5. This paper uses Bootstrap Validation (BV), Add Random Common Cause (ARCC), and Data Subsets Validation (DSV) to assess the reliability of the Average Treatment Effect (ATE), ignoring its effectiveness and universality. Could you utilize more widely used evaluation metrics to prove them of the proposed method?

---

> ### Author Response · Authors · 2023-11-22
>
> Thank you for the feedback and suggestions. We believe that we can try to prove our method in multiple fields in the future. We marked neurons with causal relationships between input layers in green, while independent neurons remained in blue. The red arrows between the input layer neurons represent the causal relationship between them. We have added text descriptions and symbol explanations for figure1 in the latest version. In the first part, we described the current work on improving the credibility of neural networks. In figure3(c), l1b is the confounding factor between l1a and lnc, and path l1alnc need to intervene the backdoor path from l1b. In addition, we supplemented the ablation experiment, and without considering the causal relationship between input layer neurons. It can be seen that our ATE slope has significantly decreased, making the analysis of neural network credibility more difficult. We apologize for not clarifying all questions given the limited space and many reviews.

---

### Meta-Review · Area_Chair_MLAT · 2023-12-09

**Metareview:**

This paper presents a method for analyzing the credibility of FNNs through a causal lens. However, the reviews collectively suggest that the paper falls short in several key areas. The motivation and theoretical underpinnings are not clearly articulated, and the causal structures and relationships are inadequately explained and justified. Moreover, the experimental validation is limited to a single dataset without comparison to state-of-the-art methods, which raises concerns about the method's robustness and generalizability. The reviewers also note that the paper needs more clarity in its presentation and definitions. Due to these significant concerns, the paper is not recommended for acceptance in its current form.

**Justification For Why Not Higher Score:**

The rebuttal is very weak and none of the reviewers were willing to increase the scores.

**Justification For Why Not Lower Score:**

n/a

---

### Decision · Program_Chairs · 2024-01-16

Reject